# Life-Course Socioeconomic Trajectories and Biological Aging: The Importance of Lifestyles and Physical Wellbeing

**DOI:** 10.3390/nu16193353

**Published:** 2024-10-02

**Authors:** Simona Esposito, Marialaura Bonaccio, Augusto Di Castelnuovo, Emilia Ruggiero, Mariarosaria Persichillo, Sara Magnacca, Amalia De Curtis, Chiara Cerletti, Maria Benedetta Donati, Giovanni de Gaetano, Licia Iacoviello, Alessandro Gialluisi

**Affiliations:** 1Department of Epidemiology and Prevention, IRCCS Neuromed, Via dell’Elettronica, 86077 Pozzilli, Isernia, Italy; simona.esposito@moli-sani.org (S.E.); marialaura.bonaccio@moli-sani.org (M.B.); dicastel@moli-sani.org (A.D.C.); emilia.ruggiero@moli-sani.org (E.R.); mariarosaria.persichillo@moli-sani.org (M.P.); sara.magnacca@moli-sani.org (S.M.); amalia.decurtis@moli-sani.org (A.D.C.); chiara.cerletti@moli-sani.org (C.C.); mbdonati@moli-sani.org (M.B.D.); giovanni.degaetano@moli-sani.org (G.d.G.); alessandro.gialluisi@gmail.com (A.G.); 2Department of Medicine and Surgery, LUM University, 70010 Casamassima, Bari, Italy

**Keywords:** biological aging, socioeconomic trajectories, pro-inflammatory lifestyles, nutrition, physical wellbeing, quality of life

## Abstract

Background/Objectives: Studies investigating the associations between life-course socioeconomic status (SES) and biological aging (the difference between biological and chronological age, Δage) have mostly been focused on epigenetic clocks and on a limited number of mediators. The aim of this study was to investigate this relationship using a blood-based aging clock, as well as the potential mediation of different factors including lifestyles or their proxies and physical and mental wellbeing. Methods: A deep-learning aging clock based on 36 blood markers was deployed, in a large Italian population cohort: the Moli-sani study (N = 4772; ≥35 years; 48% men). SES was defined as an eight-level trajectory over the life course, which was tested with Δage in linear models incrementally adjusted for age, sex, and prevalent health conditions. Moreover, the proportion of associations explained by diverse potential mediators, including diet, smoking, physical activity, alcohol, body mass index (BMI), and physical and mental quality of life (QoL) was estimated. Results: Compared to participants with a stably high SES, those showing an educational and financial downward trajectory were older than their CA (β (95%CI) = 1.28 (0.73–1.83) years), as were those with a stably low SES (0.75 (0.25–01.25) years). These associations were largely explained by the tested mediators (overall proportion: 36.2% and 66.3%, respectively), prominently by physical QoL (20.7% and 41.0%), BMI (16.8% and 34.3%), lifestyle (10.6% and 24.6%), and dietary inflammatory score (5.3% and 9.2%). Conclusions: These findings indicate that life-course socioeconomic inequalities are associated with accelerated biological aging, suggesting physical wellbeing and pro-inflammatory lifestyles as potential public health targets to slow down this process in susceptible socioeconomic strata of the population.

## 1. Introduction

Aging trajectories, characterized by a progressive decline in physiological functions [1], can be influenced by environmental (i.e., non-genetic) factors, e.g., lifestyle factors like smoking, dietary habits, or their proxies (e.g., body mass index, BMI) [2,3]. While these represent “easily” modifiable risk factors to promote healthy aging (or, in other words, to slow down the biological aging process), there are also non-modifiable (genetic) factors influencing the aging processes [4]. Other “hard to modify” factors like socioeconomic status (SES)—as measured by social and economic standing, educational attainment, household income, and occupation—have been reported to be associated with unfavorable health outcomes and worse aging trajectories [5,6,7].

Among SES exposures, socioeconomic trajectories act as a critical determinant of disease risk, since the influence of socioeconomic status permeates daily life, impacting on access to resources, opportunities and, subsequently, on health outcomes [8,9]. Indeed, a higher socioeconomic status has been linked to protective factors like access to healthcare [10], education [11], and a favorable psychosocial environment [12], which contribute to a deceleration of the aging process [12]. Conversely, chronic exposure to socioeconomic adversity and stressful environments—especially in early life—has been associated with dysregulated biological and behavioral functioning [13], as well as with disrupted mental [14] and functional health [15].

Life-course socioeconomic conditions have been reported to predict cognitive performance, with SES disadvantage being associated with lower performance across several domains including processing speed, verbal fluency, and memory [11]. In line with this evidence, neuroimaging endophenotypes of cognitive skills were also linked with SES trajectories, with upward SES mobility being associated with a larger hippocampal volume [16] and downward mobility being associated with advanced white matter aging (greater mean diffusivity as measured in magnetic resonance imaging). Interestingly, mean diffusivity was also reported to partially mediate the association between household income and cognitive performance [17].

Previous studies investigated the association between SES and biological aging, a phenomenon by which the actual underlying age of an organism (i.e., biological age) ages at a faster pace compared to chronological age, which likely results in an earlier onset of age-related chronic conditions and is measured through indices known as “aging clocks” [4]. These studies reported positive associations of accelerated biological aging with low adult SES [3,12,18]. Prominently, a low social position in early life was associated with epigenetic aging acceleration in adulthood [12,19,20,21], possibly through the dysregulation of stress response systems [20,21]. Consistently, children from socioeconomically disadvantaged conditions show a faster pace of DNA methylation (DNAm) aging in different populations [22]. Moreover, a positive DNAm age acceleration, measured through different epigenetic clocks, was associated with disadvantaged childhood social class and SES mobility [21].

Despite this promising evidence, studies investigating the relationship between life-course SES trajectories and biological aging remain scarce and are mostly focused on epigenetic clocks, with partly contrasting results [18,20,23,24,25]. Moreover, the pathways potentially mediating this association remain largely underinvestigated, these studies having mostly analyzed lifestyle factors like smoking, alcohol drinking, and adiposity measures [20,24], while neglecting the role of diet, physical and mental quality of life. The aim of this work was to clarify these aspects by testing (i) the association of different SES trajectories with a blood-based marker of biological aging in an Italian population cohort, and (ii) the potential mediation of different pathways in this association, including lifestyle and dietary pro-inflammatory habits (smoking, heavy alcohol drinking, sedentary lifestyle and unhealthy diet)), or their proxies (BMI), as well as physical and mental wellbeing.

## 2. Methods

### 2.1. Population of Study

All analyses were carried out within the Moli-sani study, a large population-based cohort of adult Italians (N = 24,325; ≥35 years; 48.11% men) living in Molise, Central Italy.

This cohort study was designed to investigate the influence of genetic and environmental factors on the onset of cardiovascular, cerebrovascular, and tumor diseases. Upon baseline recruitment (between 2005 and 2010), information on sociodemographic factors, lifestyles, and clinical variables was obtained by interviewer-administered questionnaires. Moreover, blood and urine samples were collected for the quantification of diverse circulating markers, including both cell counts and parameters and biochemical tests (see Appendix A and [3] for details). Exclusion criteria were pregnancy at the time of recruitment, disturbances in mental health or decision-making impairments, current poly-traumas or coma, and refusal to sign the informed consent form. The response rate to recruitment invitation was 70%. Participants who refused to participate were older and had a higher prevalence of chronic health conditions than those accepting the offer to participate [26,27]. Additional details of the study design are available elsewhere [28]. The Moli-sani study was approved by the Ethical Committee of the Catholic University of Rome, and all the participants provided written informed consent.

### 2.2. Outcome: Blood-Based Biological Age Based on Deep Learning

After proper quality control—removing out-of-range values of cell counts and collinear variables (cholesterol, plateletcrit, hematocrit, and mean corpuscular hemoglobin)—missing data were imputed through a k-nearest neighbor algorithm [29]. The remaining 36 features were used to estimate a systemic measure of biological age (BA) in a population free of participants reporting non-Italian ancestry and a non-faster status at the time of blood draw (23,858 participants; 12,346 women; mean (SD) age = 55.9 (12.0) years) [3]. This was accomplished through a Deep Neural Network (DNN) algorithm using circulating biomarkers, recruiting center and sex as input features, and the chronological age (CA) of each participant as a label, as described in detail elsewhere [3]. The DNN was trained over 1000 epochs, in a random 80% of the sample, using a batch size of 32—so as to minimize the loss (Mean Squared Error between BA and CA) function and avoid overfitting. Then, the optimized algorithm was used to estimate BA in the remaining 20% of the sample—which actually represents the population analyzed in the present manuscript (N = 4772) (Appendix A)—and to evaluate its accuracy (Mean Absolute Error between BA and CA = 6.0 years; r = 0.76; R^2^ = 0.57), which was comparable to previous studies in the field [30,31]. The resulting (BA) measure was then used to compute an estimate of biological aging—or Δage, defined as the difference between BA and CA—which represented the outcome measure in downstream analyses in the present manuscript. Positive values of Δage suggest an accelerated (unhealthy) biological aging and negative values indicate a decelerated (healthy) biological aging [4].

### 2.3. Exposure: Socioeconomic Indicators and Computation of SES Trajectories

Self-reported socioeconomic information was assessed at baseline (2005–2010) through a structured questionnaire administered by trained personnel. SES trajectories were computed by using three SES factors, each measured at three different time points: childhood SES at 8 years of age; educational attainment; SES during adulthood, as previously performed in this cohort [32,33].

Childhood SES at 8 years of age was investigated regarding (i) housing tenure (rented, 1 dwelling ownership, >1 dwelling ownership); (ii) access to hot water; (iii) number of rooms available in the house and number of persons living in the house. The latter two measures were used to calculate an overcrowding index (the lower, the worse), using 0.6 as the cut-off (median) value in the population. Each individual SES factor at childhood was coded as 0–1 to generate a score ranging from 0 to 3. Childhood SES was defined as this combined score being <2 (low SES) or ≥2 (high SES) [5,31].

Education was based on the highest qualification attained and was categorized as low (i.e., participants reporting up to lower secondary school) or high (i.e., secondary school or higher). SES in adulthood was measured on a six-point scale rating the following SES indicators: (a) housing tenure (0, 1, and 2 points assigned to rented, 1 dwelling ownership, and >1 dwelling ownership, respectively); (b) occupational social class (3, 2, 1, and 0 points assigned to professional/managerial; skilled non-manual occupations; skilled manual; and partly skilled, unskilled, and unclassified subjects, respectively); (c) overcrowding was obtained from the ratio between the number of rooms available in the house and the number of persons living in the household, and overcrowding was defined when the ratio was <1 (population-specific median; 0 point). Finally, low/high adult SES was defined as being below (score ≤ 3) or above (score > 3) the median of the population, respectively.

Then, dichotomized low/high SES indicators at three time points were created (SES at childhood, educational level, and adult material SES) to identify 2 × 2 × 2 = 8-level life-course trajectories, ranging from “stably high” (i.e., high childhood SES + high education +high adult SES) to “stably low” (i.e., low childhood SES + low education + low adult SES) [5].

### 2.4. Statistical Analyses

All the analyses were carried out in R v4.0.4 (available online: https://www.r-project.org/ (accessed on 15 February 2021)) or in SAS/STAT, version 9.4 (SAS Institute Inc., Cary, NC, USA). Missing data in the analyzed population (Table 1) were imputed through a k-nearest neighbor (knn) approach, using the kNN() function (k = 10) of the VIM package [29].

To investigate the association between SES trajectories and biological aging, generalized linear models incrementally adjusted for (i) CA, sex, and (ii) prevalent chronic health conditions like cardiovascular disease (CVD), cancer, type 2 diabetes, hypertension, and hyperlipidemia were built, using the stable high trajectory as a reference class. To investigate the potential moderation effects of sex on this association, the latter model was further enriched for a sex-by-SES trajectory interaction term. Moreover, sex-stratified analyses were carried out to increase the interpretability of the results. Education was not included as a covariate in the model since it was used to define the SES trajectories, nor were lifestyles, which were tested as potential mediators of the association modeled (see below).

### 2.5. Mediation Analysis

The different factors which are associated with SES and that are known to influence healthy/biological aging [3,30,31] were investigated to assess whether they could explain at least part of the association detected between SES trajectories and Δage. These included lifestyles, adherence to a Mediterranean diet (MDS), dietary inflammation score (DIS), lifestyle inflammation score (LIS; Appendix A) [34], smoking and drinking habits, leisure-time physical activity levels, and BMI—as well as physical and mental quality of life (QoL), as assessed through the validated Italian version of the self-administered Short Form 36 (SF-36) test [35]. More details about the putative mediators tested are reported below. To this end, a mediation analysis using the CMAverse package was performed [36]. This uses a counterfactual approach, which is more robust than other traditional approaches against mediator–outcome and other types of confounding, potentially accounting for exposure–mediator interaction [37]. First, a preliminary analysis to test the assumption of exposure–mediator interactions was carried out, over 100 bootstrap samples with replacement (cmest function, regression-based method, option EMint = T). Since no significant interactions were detected in this preliminary analysis (*p* > 0.05), the total, direct, and indirect effects and their confidence intervals were estimated in fully adjusted generalized linear models (Model 2) under this assumption, for each putative mediator separately and for all the analyzed mediators jointly, over 1000 bootstraps. The proportion of association explained by each mediator was calculated as Direct Effect × (Indirect Effect − 1)/(Total Effect − 1), and proportions with a *p*-value < 0.05 were considered significant.

### 2.6. Definition of Covariates and Potential Mediators

For educational attainment, subjects were divided into four categories, based on their education level completed: primary, lower secondary, upper secondary and post-secondary.

Prevalent diabetes, hypertension, and hyperlipidemia were defined as dichotomous variables (Yes/No), based on the reported and verified use of specific drugs for the treatment of these disorders. Prevalent cardiovascular disease (CVD) and cancer status were initially classified into subjects with no medical history of the disease, subjects reporting the disease with medical documentation or reporting and demonstrating the use of specific drugs (in the case of CVD), and those reporting the disease without medical documentation. The latter two classes were merged into a single class for the purpose of the present analysis.

Height and weight were measured for each participant and BMI was calculated as kg/m^2^. Waist circumference (cm) was measured in the middle between the 12th rib and the iliac crest, while hip circumference (cm) was measured around the buttocks.

The smoking status of the participants was divided into three categories based on their cigarette smoking habits: smokers, previous smokers (i.e., subjects who quit at least one year before the interview), and non-smokers. Leisure-time physical activity was assessed through a structured questionnaire (including questions on sport participation, walking, and gardening) and expressed as daily energy expenditure in metabolic equivalent task-hours (MET-h/day) [38].

Food intake was assessed through the validated Italian EPIC food frequency questionnaire [39]. The EPIC questionnaire allowed us to compute the daily energy intake for the subjects assessed (Kcal/day), as well as alcohol-consumption habits, along with a few additional questions (see [39] for details).

Based on this information, drinking status was instead classified into five categories: lifetime abstainers, former drinkers, current drinkers of 0–48.0 (moderate drinkers), >48 g/day (heavy drinkers), and non-responders for people who chose not to answer about alcohol intake, as specified elsewhere [40]. MDS (adherence to a Mediterranean diet score) was determined through the score developed by Trichopoulou et al. [41], assigning 1 point to healthy foods and 0 points to detrimental ones and defining a score ranging from 0 to 9 (the latter reflecting maximal adherence).

Physical and mental wellbeing were assessed through the validated Italian version of the self-administered SF-36 test (SF36), assessing health-related quality of life (QoL) [35,42]. The questionnaire contains 36 items measuring 8 multi-item parameters of health status and covering physical (physical functioning, role limitations due to physical health problems, bodily pain, general health perceptions) and mental domains (vitality, social functioning, role limitations due to emotional problems and mental health). For each domain, a score ranking from 0 (worst health) to 100 (best health) was calculated as the weighted sum of the questions relevant to that domain. To obtain the global scores for each component (mental and physical), relevant domains are first transformed into a *z*-score (assigning equal weight to each item), then the resulting z-scores are combined into a global z-score through a weighted mean, using weights resulting from a principal component analysis. Finally, the global scores are standardized to a normal distribution with mean = 50 and SD = 10 [43].

DIS (dietary inflammation score) and LIS (lifestyle inflammatory score) were calculated using the method described by Byrd et al. [34]. For DIS, 19 food groups (18 whole foods and beverages and 1 composite micronutrient supplement group) were selected a priori based on biological plausibility and previous literature (Appendix A). The DIS components were acquired from FFQ used in the cohort [39], and weights were developed assessing the strengths of the multivariable-adjusted associations of each individual component with a panel of circulating inflammatory biomarkers—including high-sensitivity C-reactive protein and interleukins 6, 8, and 10—as computed in [34].

LIS included four components: smoking status, physical activity, alcohol intake, and BMI, with weights determined as above. Because the weights were developed based on cross-sectional exposure–biomarker associations, for the purpose of LIS construction smoking was categorized as “current” or “former/never,” height and weight were measured and BMI was calculated as kg/m^2^, and leisure-time physical activity was assessed by an interviewer-administered structured questionnaire and expressed as daily energy expenditure in metabolic equivalent task-hours (MET-h/d) for sport, walking, and gardening [44,45,46]. For the same reason, heavy alcohol consumption was defined as >1 or >2 drinks (>14 or >28 g of ethanol, respectively)/day for women and men, respectively, while any other intake lower than these amounts were classified as moderate consumption. Individual DIS and LIS scores were then calculated as the sum of their weighted components.

## 3. Results

The analyzed population consisted of 4772 participants for whom a BA—hence, Δage—measure was computed through the DNN algorithm [3] (Table 1). Men represented 48.2% of the analyzed sample, and the mean (SD) CA and Δage were 55.9 y (11.9) and −0.89 (7.8) years, respectively. The most frequent SES trajectory was the stably low SES (27.5% of the analyzed cohort), followed by the stably high SES (18.4%) and the educational and material downward trajectory (14.4%). The least represented trajectories were mere changes in the education level during the life-course (4.1% and 5.8% for educational downward and upward trajectories, respectively). These figures are in line with the rest of the Moli-sani cohort (Appendix A, *p* = 0.73), due to the randomness of the test population selected for downstream analyses of Δage (see [3] for details).

Generalized linear models revealed significant associations between socioeconomic trajectories and biological aging (Figure 1). Indeed, those showing a decrease in their educational and financial condition during life were on average older than their CA (β (95%CI) = 1.28 (0.73–1.83) years, *p* = 5.5 × 10^−6^), compared to participants with stably high conditions, while those with a stably low SES showed a smaller but still significant positive association (0.75 (0.25–01.25) years, *p* = 0.003). The associations detected were substantially stable across the incrementally adjusted models, while no association was found for all the other trajectories (Table 2).

The sex-by-SES trajectory interaction terms did not show any significant association, and only the interaction between men and a stably low SES trajectory showed a marginal trend of association with Δage (*p* for interaction = 0.1; Appendix A), which was confirmed in sex-stratified analyses (β (CI) = 0.35 (−0.39 to 1.08) in men vs. 1.02 (0.34 to 1.70) in women; (Appendix A)). Concordant associations for the educational and material downward trajectory were detected across the two sexes (β (CI) = 1.42 (0.61 to 2.22) in men vs. 1.08 (0.31 to 1.84) in women).

The figure shows regression coefficients (β) with 95% confidence intervals (95%CIs) obtained from a model adjusted for age, sex, and prevalent health conditions (CVD, cancer, diabetes, hypertension, and hyperlipidemia), using a stably high SES (i.e., high SES in childhood, high educational level, and high SES in adulthood) as the reference class.

### Analysis of Potential Mediators

The results of the mediation analysis for the associations detected between the education and material downward and the stably low SES trajectories and Δage are reported in Table 3a,b. Among the single putative mediators tested, the largest proportion of associations was explained by the SF36 physical component (20.7 (11.7; 40.3)% for the association with the educational and material downward and 41.0 (19.7; 121.7)% for the association with the stably low SES trajectory), BMI (16.8 (9.7; 32.1)% and 34.3 (17.7; 89.8)%) and LIS (10.6 (5.4; 20.0)% and 24.6 (12.2; 70.1)%, respectively). Other significant mediations were observed for the DIS (5.3 (2.1; 11.7)% and 9.2 (3.4; 28.2)%) and for the MDS (2.7 (0.3; 6.5)% and 6.2 (1.7; 21.0)%). Overall, the putative mediators tested explained 36.2 (20.4; 67.0)% and 66.3 (34.6; 212.6)% of the associations of Δage with the educational and material downward and with the stably low SES trajectory, respectively. All other putative mediators tested (smoking, alcohol drinking, physical activity, and the SF36 mental component) did not show significant mediations (Table 3a,b). A further analysis of the mediators tested revealed differential distributions across the different SES trajectories (Table 4). Prominently, the stable low and the educational and material downward trajectories showed consistently higher BMI, LIS, and DIS scores, and lower MDS and physical wellbeing, compared to the stably high SES stratum (reference).

## 4. Discussion

In the present study, the link between life-course socioeconomic trajectories and an artificial intelligence aging clock based on blood biomarkers, with the potential to tag different aging domains in the organism, was investigated. An accelerated biological aging was detected for individuals experiencing an educational and material deterioration of their socioeconomic conditions between childhood and adulthood and for those who experienced stably low SES in the same period, compared to subjects who lived in stably high socioeconomic conditions. This finding suggests that biological aging attributable to life-course socioeconomic trajectories may be heavily affected by SES in adult life, rather than childhood, at least for those individuals experiencing a low socioeconomic position in adulthood. This evidence is generally in line with other previous studies in the field, which analyzed epigenetic (DNA methylation) age [18,20,23,24,25], although some differences occur among these works, also due to the different construction of SES exposures. Indeed, an association analysis between single measures of socioeconomic position (both from childhood and adulthood) and epigenetic clock acceleration revealed significant positive associations, which were more pronounced for early-life exposures for first-generation (Hannum and Horvath) clocks and for adult life exposures for second-generation (DNAm PhenoAge and GrimAge) clocks [23]. Other studies found that a low early-life SES was associated with increased DNAm age acceleration [20,21,25], regardless of SES in adult life [24], or even no associations between life-course SES trajectories and epigenetic age acceleration [18], in line with the previous lack of findings on childhood and adulthood SES with epigenetic aging [47]. This discrepancy may be explained by the different classification of SES exposures or even of trajectories, based often on a single indicator like professional occupation or education, rather than a combination of these with other aspects like housing and financial position. Conversely, a family-based study reported accelerated epigenetic aging for individuals experiencing unfavorable socioeconomic trajectories, prominently for the stably low SES trajectory, which was similar for all the aging clocks tested, with an effect size comparable to that observed in the present study [48]. Moreover, our findings are concordant with one of the largest and most robust studies in the field, assessing physiological aging and the pace of aging—based on blood chemistry, anthropometric, and blood-pressure measurements—in a Swiss population cohort (N ≥ 5000), over three time points and 11 years of follow-up [49]. This revealed that participants who experienced disadvantaged socioeconomic conditions during the entire lifespan exhibited an increased physiological aging at baseline and aged 10% faster than those who experienced consistently advantaged SES, with the effect size of the associations being larger for adulthood SES than for childhood SES. Of interest, a broad analysis of potential mediators revealed that lifestyle factors or their proxies and mental and physical wellbeing explained a notable overall proportion of the above-mentioned associations, both for the association with the downward trajectory (~37%) and even more so for the stably low trajectory (~66%). This effect was driven by physical wellbeing (explaining ~21% and ~41% of the above-mentioned associations), followed by BMI (~17% and ~34%), pro-inflammatory lifestyles (~11% and ~25%) and diet (~5% and ~9% for pro-inflammatory dietary score and ~3% and ~6% for Mediterranean diet score). Although to our knowledge no previous study has presented a broad mediation analysis of lifestyles and nutritional patterns in the link between SES trajectories and biological aging, this evidence is partly consistent with previous works which analyzed the mediation role of harmful lifestyles—particularly smoking, alcohol consumption, and sedentary behavior—and/or their proxies (BMI) in the link between SES exposures and epigenetic aging [12,48,49]. Prominently, our estimates are highly consistent with those by Petrovic et al. [48], reporting that between 31% and 89% of the association between adult SES exposure and different DNAm aging clocks was mediated by detrimental lifestyles or proxies (smoking, alcohol consumption, sedentary behavior, and BMI). On the contrary, these factors did not explain a significant proportion of the relationship with childhood SES [48]. Schmitz and colleagues [12] found significant associations between a cumulative index of adult socioeconomic disadvantage and accelerated DNAm aging measured through different clocks, which were only partially mediated by smoking, alcohol consumption, and obesity, suggesting that differences in health behaviors alone could not explain the SES gradient in epigenetic ageing [12]. Schrempft et al. [11], when analyzing the relationship between the pace of aging and different SES (both childhood/adulthood and life-course) exposures, found limited but consistent evidence of attenuation of the relationship after adding health behaviors like smoking, physical inactivity, and alcohol consumption. Overall, these findings are all in line with the hypothesis that the association of educational attainment and epigenetic aging may be mediated by maternal smoking during pregnancy and smoking during adulthood [50], although this is not concordant with the lack of significant mediations observed for smoking in the present study. Indeed, less consistent evidence with these findings was reported in other studies. In 1099 adults from the UK Household Longitudinal Study (28–98 years), smoking, adiposity, and alcohol consumption did not heavily affect the association between life-course SES indicators and DNAm (Hannum and Horvath) age [20]. Similar observations were made by Austin and colleagues [24] in the relationship between early-life SES and epigenetic age acceleration, where a negligible attenuation of the association was observed when adding smoking, physical activity, and waist circumference to the models. In a multi-cohort study by Fiorito and colleagues [19], multiple regression models of epigenetic aging markers vs. different predictors including education level and other lifestyle-related risk factors revealed significant associations for both SES and lifestyle exposures, although formal mediation analyses were not carried out. Specifically, the effect of low education on epigenetic aging was only partially attenuated after adding lifestyles to the models and was comparable with those of other lifestyle-related risk factors like obesity and high alcohol intake, while smoking exhibited a stronger association [19]. Overall, it is difficult to directly compare these findings among themselves and with the observations reported in the present study, especially due to the different analytical strategies used (e.g., using childhood/adulthood SES vs. life-course trajectories, testing all the lifestyles jointly vs. testing them singularly, adopting counterfactual vs. other mediation approaches). Further independent mediation studies are warranted to clarify which health behaviors represent key mediators explaining the relationship between SES and biological aging.

In addition, despite a mediating role of psychological distress being plausible in this link [20], significant evidence of such a role for mental wellbeing was not observed in the present analysis, in line with the lack of evidence observed for depressive symptoms and perceived stress in independent studies [24,25]. A novel finding of the present study is that physical wellbeing explains a large proportion of biological aging disparities across socioeconomic trajectories, highlighting the role of quality of life in this scenario. Since to our knowledge previous studies did not test any comparable index, further replications are warranted to corroborate this hypothesis. Similarly, for the first time different dietary scores to explain the association between SES trajectories and biological aging were tested, revealing significant mediations, as previously hypothesized elsewhere [23]. Again, further replications and deeper analyses are needed to untangle which dietary components drive this mediation, so as to clarify potential mechanisms linking socioeconomic disadvantage and accelerated biological aging.

### Strengths and Limitations

This study presents several points of strength. Indeed, to our knowledge this represents the first association analysis of socioeconomic trajectories and a biological aging clock based on the application of machine-learning approaches to circulating blood markers, with the potential to index more precisely aging trajectories and to tag different domains of aging, like heart, liver, and renal functions, as well as glucose homeostasis [3]. Moreover, it represents one of the largest studies in the field and the widest analysis of potential mediators in the relationship between SES trajectories and biological aging, compared to previous works [12,25,34,48]. However, our analysis also suffers from some limitations. Due to the cross-sectional design of this study, establishing clear causality and the direction of effects among SES trajectories, biological aging, and potential mediators is challenging. Specifically, while trying to conceptualize SES trajectories as an exposure, mediators such as smoking habits, physical activity, and dietary patterns might have been established earlier in life, potentially before the SES trajectory was fully formed. This temporal ambiguity complicates the traditional mediation framework, where the exposure should precede the mediator, and both should precede the outcome. Consequently, our findings may reflect complex interdependencies rather than straightforward causal pathways. Second, the retrospective nature of some information collected (e.g., childhood SES) may cause a recall bias, although the SES variables tested are usually relatively easy to recall. Also, our cohort may not be representative of other Italian populations, since it was recruited in a restricted area in central Italy (Molise region); therefore, further replications in other Italian cohorts are warranted.

Future studies should expand the range of aging clocks tested, including organ-specific biological aging measures (e.g., heart age), so as to clarify whether SES trajectories affect differently specific domains of aging, and investigate separately differential influences of maternal vs. paternal SES, which may act as an effect modifier in the investigated relationship.

## 5. Conclusions

In conclusion, our findings highlight that socioeconomic inequalities over the life course are associated with an accelerated biological aging in adulthood. These findings suggest important implications for health policy, highlighting the need for targeted interventions to address lifelong socioeconomic disadvantage. Policies aimed at improving physical wellbeing, promoting healthier lifestyles, and reducing pro-inflammatory behaviors could play a critical role in mitigating accelerated biological aging, particularly among individuals in lower or declining socioeconomic strata. By focusing on these modifiable factors and on these socioeconomically vulnerable population groups, public health efforts could more effectively contribute to reducing SES-related health disparities in aging and improving long-term population health.

## Figures and Tables

**Figure 1 nutrients-16-03353-f001:**
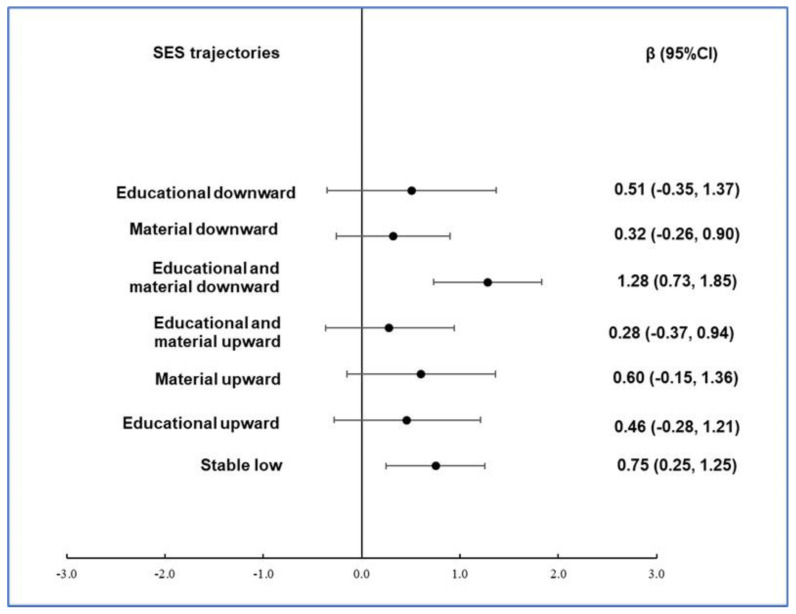
Association of Δage with SES trajectories in the Moli-sani cohort.

**Table 1 nutrients-16-03353-t001:** Characteristics of the analyzed sample from the Moli-sani study cohort (n = 4772).

Variables	N of Subjects	Mean	SD	%
Chronological age, CA (y)	4772	55.9	11.9	
Biological age, BA (y)	4772	55.0	8.7	
Δage (BA−CA)	4772	−0.89	7.8	
Sex (men)	2299	–	–	48.2
Education level				
Up to lower school	2578	–	–	54.0
Upper secondary	1632	–	–	34.2
Postsecondary education	555	–	–	11.6
Missing data	7	–	–	0.1
Housing tenure				
Rent	454	–	–	9.5
1 dwelling ownership	3884	–	–	81.4
>1 dwelling ownership	426	–	–	8.9
Missing data	8	–	–	0.2
Place of residence				
Rural	1593	–	–	33.4
Urban	3179	–	–	66.6
Body mass index (BMI)	4766	28.3	4.8	
Lifestyles				
Leisure-time physical activity (met−h/day)	4720	3.5	4.0	
Smoking status				
Non−smoker	2390	–	–	50.2
Smokers	1066	–	–	22.4
Former	1306	–	–	27.4
Missing data	10	–	–	0.2
LIS	4524	0.6	0.8	
Dietary information				
MDS	4750	4.4	1.6	
DIS	4524	−0.2	2.0	
Alcohol intake (g/day)	4752	20.3	7.3	
Cardiovascular disease				
No	4421	–	–	92.6
Yes	276	–	–	5.8
Missing data	75	–	–	1.6
Cancer				
No	4591	–	–	96.2
Yes	151	–	–	3.2
Missing data	30	–	–	0.6
Diabetes				
No	4470	–	–	93.7
Yes	234	–	–	4.9
Missing data	8	–	–	1.4
Hypertension				
No	3300	–	–	69.2
Yes	1428	–	–	29.9
Missing data	44			0.9
Hyperlipidemia				
No	4337	–	–	91.9
Yes	384	–	–	8.1
Missing data	51	–	–	1.1
Quality of life				
SF−36 physical QoL	3728	46.6	6.4	
SF−36 mental QoL	3728	46.9	10.1	

Abbreviations: MDS = Mediterranean diet score; DIS = dietary inflammatory score; LIS = lifestyle inflammatory score; QoL = quality of life.

**Table 2 nutrients-16-03353-t002:** Association of Δage with SES trajectories in incrementally adjusted models.

SES Trajectories	β ^1^ (95%CI)	β ^2^ (95%CI)
Stable high	Reference	Reference
Education downward	0.38 (−0.50 to 1.26)	0.51 (−0.35 to 1.37)
Material downward	0.29 (−0.31 to 0.89)	0.32 (−0.26 to 0.90)
**Education and material downward**	**1.37 (0.81 to 1.94)**	**1.28 (0.73 to 1.83)**
Education and material upward	0.26 (−0.41 to 0.93)	0.28 (−0.37 to 0.94)
Material upward	0.72 (−0.01 to1.50)	0.60 (−0.16 to 0.14)
Education upward	0.57 (−0.19 to 1.33)	0.46 (−0.29 to 1.21)
**Stable low**	**0.93 (0.42 to 1.44)**	**0.75 (0.25 to 1.25)**

Regression coefficients (β) with 95% confidence intervals (95%CIs) are reported for each SES trajectory, compared to the stably high SES condition (reference class). Significant associations (*p* < 0.05) are highlighted in bold. ^1^ Regression coefficients (95%CI) obtained from a model adjusted for chronological age and sex (Model 1: Δage~SEStrajectory + CA + sex). ^2^ Regression coefficients (95%CI) obtained from a model adjusted for chronological age, sex, prevalent CVD, cancer, diabetes, hypertension, and hyperlipidemia (Model 2: Δage~SEStrajectory + CA + sex + CVD + cancer + diabetes + hypertension + hyperlipidemia).

**Table 3 nutrients-16-03353-t003:** Mediation analysis for associations detected between biological aging (Δage) and SES trajectories in the Moli-sani cohort (n = 4772).

**(a)**				
**Potential Mediators**	**Total Effect** **β [95% CI]**	**Pure Natural** **Direct Effect** **β [95% CI]**	**Pure Natural** **Indirect Effect** **β [95% CI]**	**% Mediation** **[95% CI]** **(*p*-Value)**
LIS score	**1.28 [0.74; 1.78]**	**1.15 [0.63; 1.65]**	**0.14 [0.07; 0.21]**	**10.6 (5.4; 20.0)%** ***p* < 0.001**
DIS score	**1.28 [0.73; 1.78]**	**1.21 [0.65; 1.72]**	**0.07 [0.03; 0.12]**	**5.3 (2.1; 11.7)%** ***p* = 0.002**
MDS	**1.28 [0.70; 1.78]**	**1.25 [0.68; 1.75]**	**0.03 [0.003; 0.075]**	**2.7 (0.3; 6.5)%** ***p* = 0.028**
Smoke	1.28 [0.71; 1.81]	1.28 [0.72; 1.82]	−0.002 [−0.06; 0.06]	−0.2 (−6.2; 4.9)% *p* = 0.92
Physical activity	1.28 [0.74; 1.84]	1.28 [0.74; 1.84]	0.004 [−0.009; 0.022]	0.3 (−0.7; 1.9)% *p* = 0.57
Alcohol drinking	1.29 [0.73; 1.85]	1.27 [0.70; 1.84]	0.02 [0.14; 0.77]	1.2 (−8.6; 11.1)% *p* = 0.77
BMI	**1.28 [0.74; 1.82]**	**1.07 [0.52; 1.62]**	**0.22 [0.14; 0.31]**	**16.8 (9.7; 32.1)%** ***p* < 0.001**
SF36 physical	**1.28 [0.70; 1.90]**	**1.02 [0.43; 1.62]**	**0.27 [0.17; 0.38]**	**20.7 (11.7; 40.3)%** ***p* < 0.001**
SF36 mental	1.28 [0.73; 1.83]	1.29 [0.72; 1.84]	−0.003 [−0.019; 0.009]	−0.3 (−1.7; 0.1)% *p* = 0.65
ALL	**1.27 [0.75; 1.84]**	**0.81 [0.26; 1.41]**	**0.46 [0.29; 0.67]**	**36.2 (20.4; 67.0)%** ***p* < 0.001**
**(b)**				
**Potential Mediators**	**Total Effect** **β [95% CI]**	**Pure Natural** **Direct Effect** **β [95% CI]**	**Pure Natural** **Indirect Effect** **β [95% CI]**	**% Mediation** **[95% CI]** **(*p*-Value)**
LIS score	**0.75 [0.26; 1.27]**	**0.57 [0.07; 1.06]**	**0.18 [0.11; 0.27]**	**24.6 (12.2; 70.1)%** ***p* = 0.004**
DIS score	**0.75 [0.25; 1.28]**	**0.68 [0.19; 1.22]**	**0.07 [0.03; 0.12]**	**9.2 (3.4; 28.2)%** ***p* = 0.004**
MDS	**0.75 [0.22; 1.24]**	**0.70 [0.18; 1.20]**	**0.05 [0.01; 0.09]**	**6.2 (1.7; 21.0)%** ***p* = 0.010**
Smoke	0.77 [0.21; 1.22]	0.79 [0.25; 1.25]	−0.01 [−0.10; 0.02]	−1.7 (−26.7; 2.7)% *p* = 0.23
Physical activity	0.75 [0.25; 1.23]	0.74 [0.23; 1.21]	0.01 [−0.02; 0.04]	1.4 (−2.4; 8.3)% *p* = 0.50
Alcohol drinking	0.74 [0.25; 1.26]	0.76 [0.26; 1.29]	−0.02 [−0.12; 0.10]	−2.5 (−24.7; 17.5)% *p* = 0.86
BMI	**0.75 [0.27; 1.27]**	**0.49 [0.03; 1.01]**	**0.26 [0.18; 0.35]**	**34.3 (17.7; 89.8)%** ***p* = 0.002**
SF36 physical	**0.75 [0.23; 1.25]**	**0.44 [−0.10; 0.94]**	**0.31 [0.19; 0.44]**	**41.0 (19.7; 121.7)%** ***p* = 0.014**
SF36 mental	0.75 [0.24; 1.24]	0.74 [0.24; 1.24]	0.006 [−0.006; 0.025]	0.8 (−0.8; 4.2)% *p* = 0.036
ALL	**0.72 [0.23; 1.28]**	**0.25 [−0.29; 0.78]**	**0.48 [0.31; 0.70]**	**66.3 (34.6; 212.6)%** ***p* = 0.002**

The association β and the proportion of association mediated are reported for (**a**) the educational and material downward and (**b**) the stably low SES trajectory, for each potential mediator tested, along with 95% confidence intervals, as computed through the CMAverse package [32]. β represents the counterfactual effect conditional on covariates specified in the fully adjusted linear model (Model 2). *%* mediation was computed as Direct Effect × (Indirect Effect − 1)/(Total Effect − 1). Significant mediations (*p* < 0.05) are highlighted in bold. No exposure–mediator interaction was assumed, since no significant evidence of such an interaction was detected. Abbreviations: DIS = dietary inflammatory score; LIS = lifestyle inflammatory score; MDS = Mediterranean diet score; BMI = body mass index.

**Table 4 nutrients-16-03353-t004:** Characteristics of the study population by socioeconomic trajectories, in the Moli-sani cohort analyzed (n = 4772).

	SES Trajectories								
	Stable High (N = 891)	Education Downward (N = 198)	Material Downward (N = 590)	Education and Material Downward (N = 703)	Education and Material Upward (N = 426)	Material Upward (N = 289)	Education Upward (N = 282)	Stable Low (N = 1393)	*p*-Value
LIS score (mean; SD)	0.43; 0.79	0.68; 0.76	0.47; 0.74	0.63; 0.77	0.52; 0.77	0.63; 0.70	0.56; 0.77	0.71; 0.70	<0.0001
DIS score (mean; SD)	−0.37; 2.27	−0.01; 2.24	0.03; 2.13	0.20; 2.34	−0.23; 2.02	−0.18; 2.13	−0.03; 1.98	0.20; 1.98	<0.0001
MDS (mean; SD)	4.5; 1.7	4.5; 1.7	4.3; 1.6	4.3; 1.6	4.5; 1.7	4.3; 1.6	4.5; 1.7	4.2; 1.6	0.005
Smoking status (%)									<0.0001
Non-smoker	47.1	42.9	51.0	50.6	42.2	48.4	39.7	57.5	
Smoker	27.3	32.3	23.2	23.5	34.7	36.3	29.8	26.1	
Former	25.6	24.7	25.8	25.9	23.0	15.2	30.5	16.4	
Physical activity (mean; SD)	3.2; 3.4	3.9; 4.4	3.2; 3.4	3.4; 4.3	3.2; 3.2	3.7; 3.9	3.3; 3.6	3.8; 4.6	0.005
Alcohol drinking (%)									<0.0001
Non-responder	1.3	2.5	2.0	4.3	2.1	4.1	3.2	8.0	
Former drinker	4.4	6.1	2.7	4.5	4.7	7.3	2.5	5.5	
Lifetime abstainer	32.7	40.9	35.9	38.7	26.8	32.5	38.6	33.4	
Moderate	54.3	29.8	48.3	27.7	54.5	38.4	35.1	28.1	
Heavy	7.3	20.7	11.0	24.7	12.0	17.6	20.6	25.1	
BMI (mean; SD)	27.1; 4.5	28.6; 4.6	27.4; 4.6	28.8; 5.1	27.8; 4.2	28.7; 4.4	27.7; 4.4	29.2; 5.0	<0.0001
SF36 physical (mean; SD)	48.5; 4.8	46.3; 5.3	47.6; 5.1	45.1; 5.7	47.7; 5.8	45.8; 5.9	47.3; 4.7	44.6; 6.1	<0.0001
SF36 mental (mean; SD)	47.1; 8.8	47.1; 8.7	46.9; 9.4	47.3; 8.8	46.9; 10.7	47.6; 8.4	46.5; 9.6	46.5; 8.9	0.46

*p*-values were obtained using generalized linear models for continuous variables and logistic regression for categorical variables, adjusted for age and sex, while socioeconomic trajectories were modeled as predictors.

## Data Availability

The data underlying this article will be shared upon reasonable request to the corresponding author. The data are stored in an institutional repository (https://repository.neuromed.it) and access is restricted by the ethics approval and the legislation of the European Union.

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
