# Peer review of "Life-Course Socioeconomic Trajectories and Biological Aging: The Importance of Lifestyles and Physical Wellbeing"

_nutrients, 2024, doi:10.3390/nu16193353_

Round 1

Reviewer 1 Report

Comments and Suggestions for Authors

Dear Authors,

Thank you for your manuscript. The paper is well-written and makes a valuable contribution to public health. Please see my comments below.

In the abstract (line 30) and the final paragraph of the Introduction (lines 79-80), you mention "pro-inflammatory lifestyle." It would be helpful to clarify this term before using it.

Although BMI is a commonly used abbreviation, it is recommended to provide the full term when it is first introduced.

I suggest moving the description of the mediator methods from the supplementary materials to the Methods section, as this information is crucial for interpreting the results.

The results are clearly presented and thoroughly discussed.

Comments on the Quality of English Language

Minor English editing is required is accepted.

Author Response

Dear Authors,

Thank you for your manuscript. The paper is well-written and makes a valuable contribution to public health. Please see my comments below.

In the abstract (line 30) and the final paragraph of the Introduction (lines 79-80), you mention "pro-inflammatory lifestyle." It would be helpful to clarify this term before using it.

Thank you for this useful suggestion, we reported this information in the Introduction section.

Although BMI is a commonly used abbreviation, it is recommended to provide the full term when it is first introduced.

Agreed. Thanks for your suggestion.

I suggest moving the description of the mediator methods from the supplementary materials to the Methods section, as this information is crucial for interpreting the results.

Agreed. We moved the relevant paragraph as suggested.

The results are clearly presented and thoroughly discussed.

We thank the reviewer for the positive feedback.     

Reviewer 2 Report

Comments and Suggestions for Authors

The manuscript focuses on the analyses of associations among socioeconomic trajectories and biological aging, emphasizing the role of lifestyle and wellbeing characteristics of individuals. The subject is very interesting and relevant in the field of public health; however, it seems to be out of the scope of the journal Nutrients.

The Introduction and the Discussion sections of the manuscript require additional literature review to highlight the role of socioeconomic trajectories, lifestyle, and wellbeing on aging. Furthermore, the identification of literature gaps and the exploration of diet and nutrients should be incorporated into the study to highlight the connection between the paper and the scope of the journal.

The Materials and Methods should bring additional information on the Moli-sani study to support the understanding of the study design, follow-up, and variables of the survey (i.e., include an flowchart with survey participants from sample calculation to data analyses in the manuscript). The statistical analyses performed should include the detailed definition of variables and models (i.e., description of variables within the section instead of presentation in the supplementary materials, and inclusion of equations of the models).

The role of dietary characteristics of participants (especially nutrients) should be emphasized in the models, since the scope of the journal refers to nutrients.

The description of results should be carefully revised, since there are imprecisions (e.g., page 7, lines 213-214, authors indicate that "only the interaction between men and stably low SES trajectory showed a marginal trend of association with Δage"; however, according to the data shown in Table 3, education and material conditions downward trajectory was also marked as significant in bold.

Finally, the Discussion of the paper should be enriched with additional comparison with findings from other studies in the subject, in addition to exploration of potential implications of the results of the study on health policies in the country.

Comments on the Quality of English Language

The quality of English language should be improved, there are certain parts of the paper that are difficult to understand or rely on informal language rather than scientific terms.

Author Response

The manuscript focuses on the analyses of associations among socioeconomic trajectories and biological aging, emphasizing the role of lifestyle and wellbeing characteristics of individuals. The subject is very interesting and relevant in the field of public health; however, it seems to be out of the scope of the journal Nutrients.

Thank you for your comment, which allows us to clarify this aspect. The present manuscript was submitted to the special issue “The Effect of Dietary Patterns and Lifestyle on Healthy Aging”, and was preliminarily checked by the editorial office of the journal, which considered it adequate for the scope of the special issue. Indeed, one of the aims was to show how important are nutrition and other lifestyles in mediating the association between unfavorable socioeconomic trajectories and accelerated (unhealthy) biological aging, as stated in the last paragraph of the Introduction section. To this end and to better convey our aim, we modified the manuscript as suggested by this reviewer (see below).

The Introduction and the Discussion sections of the manuscript require additional literature review to highlight the role of socioeconomic trajectories, lifestyle, and wellbeing on aging. Furthermore, the identification of literature gaps and the exploration of diet and nutrients should be incorporated into the study to highlight the connection between the paper and the scope of the journal.

Thank you for your suggestion. We added the required information to the Introduction and to the Discussion section. We expanded the Discussion section focusing on the comparison with previous studies. Although we found a couple of additional studies analyzing the relationship between life-course SES conditions and biological aging (e.g. Raffington et al., 2021, doi: 10.1542/peds.2020-024406; 2023, 10.1186/s13148-023-01489-7; Lawn et al., 2018, 10.1093/hmg/ddy036), we did not find any other study specifically addressing the question whether pro-inflammatory nutritional patterns and lifestyles explain the association between unfavorable socioeconomic life-course trajectories and accelerated biological aging, therefore highlighted this literature gap in the text. This topic was partially addressed by Fiorito and colleagues (2019; doi: 10.18632/aging.101900), who however tested associations in multiple regression models including education level (used as SES indicator) and other lifestyle-related risk factors like smoking, obesity, alcohol intake, and low levels of physical activity. Moreover, this work (now added to the Discussion section) did not account for the life-course SES trajectories. Should the reviewer be aware of any further study that we may have missed, we are happy to receive feedbacks with this regard.

The Materials and Methods should bring additional information on the Moli-sani study to support the understanding of the study design, follow-up, and variables of the survey (i.e., include an flowchart with survey participants from sample calculation to data analyses in the manuscript). The statistical analyses performed should include the detailed definition of variables and models (i.e., description of variables within the section instead of presentation in the supplementary materials, and inclusion of equations of the models).

Thank you for the valuable suggestions. We have added further information about the Moli-sani study in the Methods section, and a flowchart of the analyzed sample in the supplementary file (Supplementary Figure 1). Additionally, as suggested, the definitions and descriptions of the variables used in the models have been moved from the supplementary file to the Methods section in the main text. We also better specify the equations of the linear models carried out in the caption of Table 2.

The role of dietary characteristics of participants (especially nutrients) should be emphasized in the models, since the scope of the journal refers to nutrients.

We thank the reviewer for this suggestion, we added a description of the study population by showing the distribution of the mediators tested (including diet and lifestyles) across the different socioeconomic trajectories. You can find the results of this additional analysis in Table 4.

The description of results should be carefully revised, since there are imprecisions (e.g., page 7, lines 213-214, authors indicate that "only the interaction between men and stably low SES trajectory showed a marginal trend of association with Δage"; however, according to the data shown in Table 3, education and material conditions downward trajectory was also marked as significant in bold.

We thank the reviewer for the useful observation. Indeed, Table 3 was showing the associations of both the additive and the interactive term in the regression model and only the additive term was highlighted as significant (which may actually be misleading). To avoid misunderstandings and since this information was redundant with the one reported in Table 2 (results of the main additive models), we deleted this column. Moreover, since this table was not showing any significant interactive association and in view of the introduction of a new table in the main text, we moved this table to the supplementary file (now called Table S3).

Finally, the Discussion of the paper should be enriched with additional comparison with findings from other studies in the subject, in addition to exploration of potential implications of the results of the study on health policies in the country.

We thank the Reviewer for this suggestion. We expanded the Discussion by further detailing findings from other studies – particularly related to the mediation analysis, which represents one of the main novelties of our study - and added the potential implications of the results of the present study on health policies in the Conclusions.

Reviewer 3 Report

Comments and Suggestions for Authors

The authors have a well written manuscript.  It is a novel study using learning aging clock. Technically, the paper was written well and they described thoroughly their methodology. Methodology appears robust.

Couple things

1. The minor part was for them to fix 1st person language. Don't use 1st person language.  Avoid the "we" language, several examples throughout.

2. They highlighted limitations and I don't think overstep their findings.

3. Missing a conclusion paragraph at the end.

Author Response

The authors have a well written manuscript.  It is a novel study using learning aging clock. Technically, the paper was written well and they described thoroughly their methodology. Methodology appears robust.

Couple things

  1. The minor part was for them to fix 1st person language. Don't use 1st person language. Avoid the "we" language, several examples throughout.

Thank you, we changed as suggested throughout the text.

  1. They highlighted limitations and I don't think overstep their findings.

We thank the reviewer for the positive feedback.     

  1. Missing a conclusion paragraph at the end.

We added the conclusion paragraph as suggested, thank you.

Round 2

Reviewer 2 Report

Comments and Suggestions for Authors

The manuscript was revised according to the suggestions indicated in my review, improving the quality and understanding of the study.